# A multi-stable deployable quadrifilar helix antenna with radiation reconfigurability for disaster-prone areas

Rosette Maria Bichara[1], Joseph Costantine [1] ✉, Youssef Tawk[1] & Maria Sakovsky [2] ✉

In disaster-prone areas, damaged infrastructure requires impromptu communications leveraging lightweight and adaptive antennas. Accordingly, we introduce a bi-stable deployable quadrifilar helix antenna that passively reconfigures its radiation characteristics in terms of pattern and polarization. The proposed structure is composed of counter-rotating helical strips, connected by rotational joints to allow a simultaneous change in the helix height and radius. Each helical strip is composed of a fiber-reinforced composite material to achieve two stable deployed states that are self-locking. The reconfiguration between an almost omnidirectional pattern and a circularly polarized directive pattern enables the antenna to be suitable for both terrestrial and satellite communication within the L-band. More specifically, the presented design in infrastructure-less areas achieves satellite localization with directive circularly polarized waves and point-to-point terrestrial connectivity with an almost omnidirectional state. Hence, we present a portable, agile, and passively reconfigured antenna solution for low-infrastructure areas.

Natural disasters devastate regions and eliminate available communication infrastructure, rendering rescue efforts extremely challenging[1]. Furthermore, underdeveloped regions already struggle with diminished infrastructure and have an underserved population that lacks basic communication needs, hindering humanitarian efforts. Antennas currently deployed in disaster-stricken regions consist of large dish reflectors with heavy weight and are not easily deployed or moved[2–4]. In addition, communication needs in regions with diminished infrastructure necessitate agility in antenna performance that enables terrestrial and satellite communications alike – a feature that existing solutions lack. Hence, there is a need to develop antenna systems that can deploy on demand and reconfigure their performance to satisfy arising circumstances while being lightweight and requiring minimum power.

Various types of deployable antennas have been investigated in literature including helical antennas, conical log spirals, quadrifilar helix antennas, origami and kirigami structures, and many other topologies[5–14]. While the deployable aspect of these antennas constitutes a major component of this work, the reconfiguration of these structures is another component that must be tackled. Reconfigurable antennas can switch their operational frequencies on demand, alternate their radiation patterns, or achieve polarization reconfiguration[15–20]. Reconfiguration techniques range from the use of electrical switches[21,22] to smart materials[18,23,24] or to mechanical reconfiguration of conductors[25,26]. However, the merging of both deployment and reconfiguration mechanisms necessitates the use of an actuation system. Previous work has suggested traditional actuators such as motors or pneumatic actuators[7,26,27] as well as smart material actuation[18,19,23] that must be continuously powered to achieve a certain antenna topology. To address this challenge, the reconfiguration mechanism in this work is made completely passive by relying on a manual reconfiguration process, thus removing the need for dc power

[1]Department of Electrical and Computer Engineering, Maroun Semaan Faculty of Engineering and Architecture, American University of Beirut, Beirut 1107 2020, Lebanon. [2]Department of Aeronautics and Astronautics, Stanford University, Stanford CA94305, USA. ✉e-mail: jcostantine@ieee.org; sakovsky@stanford.edu

and external biasing. Such a feature constitutes a zero-power approach in implementing agile communications in diminished infrastructure areas. This work addresses the two challenges associated with a user-centric reconfigurable deployable antenna. First, it ensures that the antenna can be repeatedly reconfigured into the desired geometries, which are all held passively. Second, coupling the passive geometries to the desired electromagnetic reconfiguration of the antenna itself.

The proposed bi-stable deployable and reconfigurable quadrifilar helix antenna (QHA) for areas with diminished infrastructure is depicted in Fig. 1. The antenna[28] deploys and offers two stable geometric configurations to achieve polarization and radiation pattern reconfiguration, leading to on-demand terrestrial and satellite communications. The stable geometries correspond to the minima of the stored strain energy in the structure. Therefore, the antenna will return to one of these two states automatically, even when perturbed away from them. Specifically, the antenna uses a helical lattice composed of counter-rotating helical strips connected by rotational joints allowing a simultaneous change in the helix height and radius. The strips are composed of a fiber-reinforced composite material whose anisotropic mechanical properties result in a lattice with two stable deployed states that are self-locking. The appropriate transition between the two states is ensured through a sliding feed topology. Such a feeding mechanism slides within the ground plane of the antenna and accompanies its geometrical deployment while maintaining the antenna operation within the L-band (1-1.2 GHz). The resulting structure with its structurally adaptive feeding is lightweight, with a stable frequency band response and reconfigurable radiation. In addition, the antenna is user-centric, and as a result, requires no Dc input power for reconfiguration while still ensuring repeatable shape change due to bi-stability. All these features define the major aspects of the proposed design and address the identified challenges of enabling agile communications in diminished infrastructure areas.

## Results
### Reconfigurable fiber-reinforced polymer antenna
The proposed structure is composed of four conductive strips wound counter-clockwise and four dielectric strips wound clockwise. The strips are interconnected using non-conductive rotational joints as

shown in Fig. 2a. The joints enable the whole structure to simultaneously vary its height, $h$, and radius, $R$. The reconfigurable deployable antenna concept proposed here is illustrated in Fig. 2b. The proposed design is based on a quadrifilar helix antenna where the dielectric strips act as a structural support for the conductive helices and enable reconfiguration between states. The deformation of the helical structure is enabled through the bending and twisting of the strips and any length changes are relatively small[29]. As a result, the height of the helix, $h$, is related to its radius, $R$, using (1)[29], where $l$ is the length for each of the strips and $N$ is the number of turns.

$$h = 2\pi N \sqrt{\left(\frac{l}{2\pi N}\right)^2 - R^2} \tag{1}$$

Reconfiguration is executed by applying a vertical force on the tips of the strips that make up the structure as depicted in Fig. 2b. This force enables the eight strips to slide along the structure's ground plane by following the four arrows that are parallel to the ground plane. Applying a force in the (−z) direction reduces the height of the helix and increases its radius while applying a force in the (+z) direction reduces the radius of the helix and increases its height. The relationship between the helix height and radius for the length of the strip ($l = 188.5$ mm) and the number of turns ($N = 0.5$ turns) is presented in Fig. 2c.

In the remainder of this section, we identify the particular combinations of radius and height (i.e., helix geometries) that result in the desired radiation pattern and polarization reconfiguration. Accordingly, we then present a material and structural concept to ensure the selected helix geometries correspond to the minima of the deformation strain energy of the structure – making these geometries stable states that require no external energy to be maintained.

### Two distinct operational states
The electromagnetic response of the proposed QHA was assessed by simulations for various radius and height combinations as per Fig. 2c. These different geometries were simulated using Ansys High-Frequency Structure Simulator (HFSS)[30], discussed in the Methods

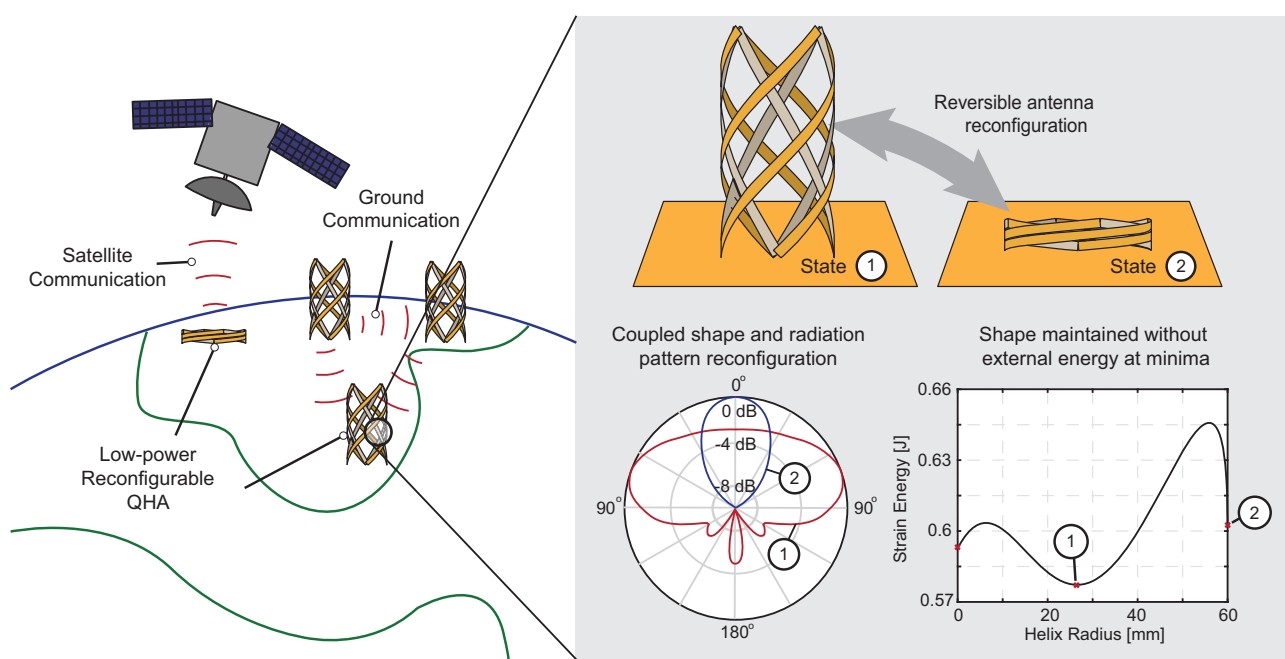

**Fig. 1 | Operational concept of the multi-stable deployable and reconfigurable quadrifilar helix antenna for Disaster-Prone Areas.** Overview of the coupled structural and electromagnetic reconfiguration used by the proposed low-power quadrifilar helix antenna for low-infrastructure areas.

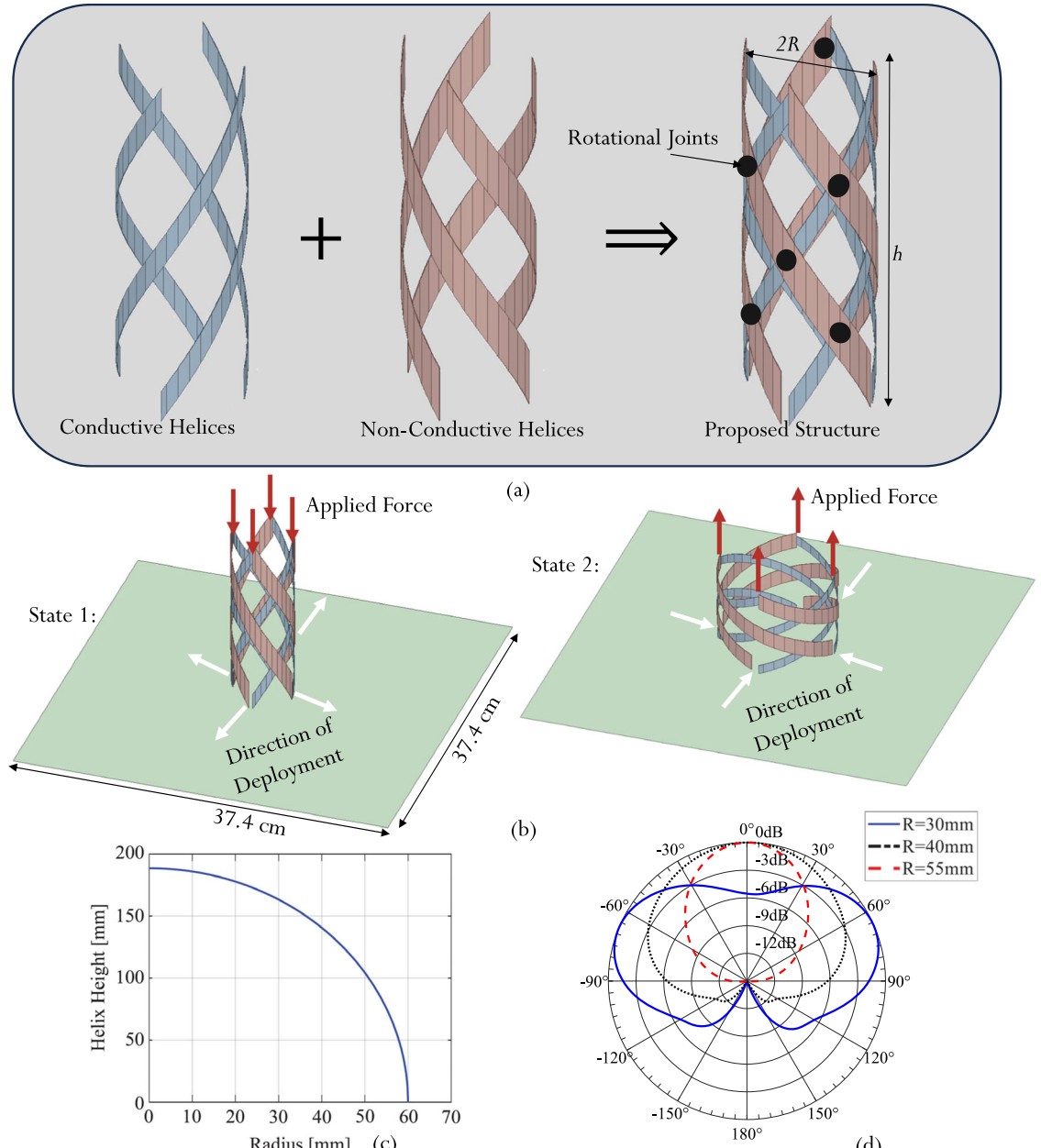

**Fig. 2 | Design of a bi-stable reconfigurable QHA. a** Antenna structure and topology. **b** Antenna reconfiguration concept and mechanism. **c** Relationship between helix height and radius for the length of the strip ($l = 188.5$ mm) and the total number of turns ($N = 0.5$). **d** Radiation pattern variation as a function of radius at 1.1 GHz.

section. Supplementary Table 1 provides the detailed antenna performance metrics as a function of the helix geometry. It was found that for a radius ($R$) less than 25 mm, the antenna structure is not well matched. For 25 mm $<R<$ 35 mm, the antenna exhibits a close to omnidirectional pattern. A radius between 45 mm $<R<$ 55 mm allows the antenna to exhibit a directional pattern with circularly polarized radiation behavior. For $R > 55$ mm, the antenna design is no longer matched.

The simulated and normalized radiation patterns representing these different operating regimes are shown for three different radii in Fig. 2d. It is important to note that for a radius between 35 mm and 45 mm, the antenna starts its operational transition from an omnidirectional pattern to a directional one. Since the conductive strip length remains constant throughout the reconfiguration, the operating frequency remains within the antenna bandwidth between 1 GHz and 1.2 GHz. Accordingly, two distinct antenna operating states are

identified. These states reconfigure the radiation characteristics of the antenna. State 1 is characterized by a radius of $R = 30$ mm and an almost omnidirectional radiation pattern. These characteristics make it suitable for terrestrial communications between various devices deployed in a low-infrastructure or disaster-stricken area. On the other hand, State 2 is characterized by a radius of around $R = 55$ mm with a directive radiation pattern and circular polarization, making it ideal for satellite communications.

The corresponding simulated magnitude of the 3D radiated electric field pattern for state 1 and state 2 are presented in Fig. 3a, b at $f = 1.1$ GHz. The 2D left- and right-handed circularly polarized gain patterns along the $Y − Z$ plane are presented for state 2 in Fig. 3c. A cross-polarization of 41.68 dB is obtained along the direction of maximum radiation at $\theta = 0°$. This confirms the circular polarization purity of the antenna in state 2.

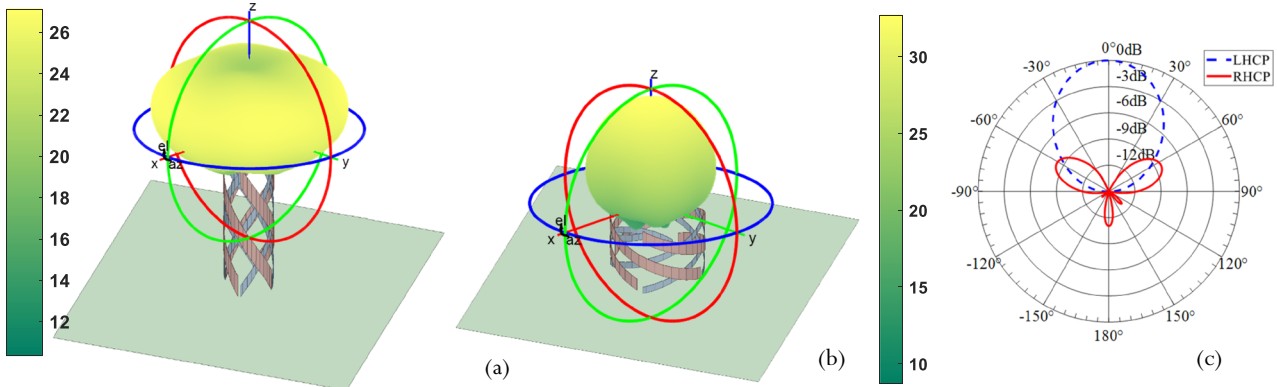

**Fig. 3 | Simulated QHA performance at 1.1 GHz in two selected operating states.** **a** Antenna in state 1 with its 3D radiated electric field magnitude in dB. **b** Antenna in state 2 with its 3D radiated electric field in dB. **c** The simulated Left Hand Circularly Polarized (LHCP) and Right Hand Circularly Polarized (RHCP) two-dimensional radiation pattern for State 2, proves that in State 2 the antenna is LHCP.

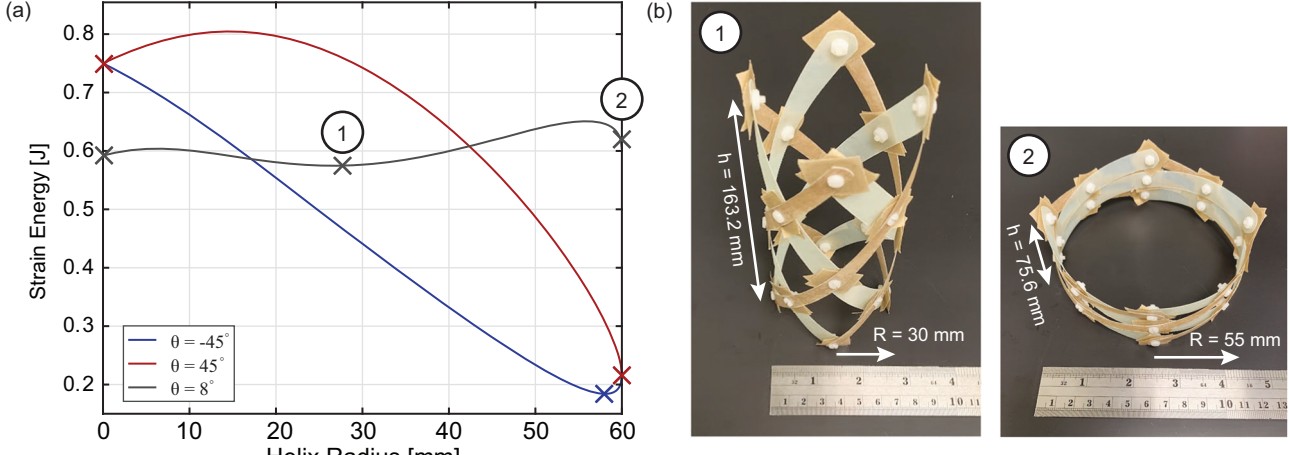

**Fig. 4 | Bi-stable structural design. a** Variation in stored deformation strain energy as a function of radius and fiber-reinforced composite material properties. **b** Fabricated antenna in states 1 and 2 and the physical dimensions of the two operating states.

## Multi-stable composite structure

Each helix is composed of initially flat strips of glass fiber and epoxy composites with an embedded conductive mesh. These strips are interconnected using rotational joints into a helical lattice as discussed in the Methods section. As the helix is reconfigured between various geometries, the strips undergo bending and twisting[31], thereby changing the strain energy stored in the structure due to the elastic deformation of the material. The stored strain energy, $U$, results in forces acting along the axis of the helix according to Castigliano's theorem[32],

$$F_a = \frac{\partial U}{\partial h} \qquad (2)$$

These forces deform the structure towards its minimum energy configuration. The energy minima are the stable points of the structure, where the restoring forces are $F_a = 0$ and the structure can maintain its geometry without external power. In the case that the force pushes the structure towards its minimum or maximum radius, there are additional stable points at these extremes. We leverage such behavior in our study to enable the low-energy, user-centric reconfiguration of the structure. The desired geometries are reliably achieved as axial forces due to the stored strain energy and will always push the structure to the minimum energy configuration. No DC power is required for actuators to maintain stable geometries.

The use of fiber-reinforced polymer (FRP) composites for the strip material has been shown in the literature to yield a rich design space for stability[31]. The stored strain energy can be predicted as a function of material properties and helix geometry using a 2D analytic model accounting for longitudinal and transverse bending as well as twisting in the strips. A detailed discussion of the design parameters is presented in the Methods section. We demonstrate that we can use the fiber orientation of the composite material relative to the strip length, $\theta$, to induce a multi-stable response. Figure 4a demonstrates the use of fiber orientation in the strips to increase the number of stable points for a helix made of glass-fiber-reinforced epoxy material. It is found that a fiber angle of 8° results in two stable points at radii corresponding to States 1 and 2, thereby successfully coupling mechanical and electromagnetic reconfiguration. The third stable point at the minimum radius is not used in practice. At this fiber angle, the FRP material has low torsional stiffness as well as high material coupling between bending and twisting deformations. The helical strips twist significantly to enable multi-stability. The stable radii are measured when the QHA is integrated with its ground plane. These agree well with the predictions in Fig. 4a with small deviations due to manufacturing imperfections, the physical width of the strips, and integration with the ground plane. The antenna is shown in its two stable states in Fig. 4b. The resulting dimensions of the antenna structure in the two stable operating states with radii of $R = 30$ mm and $R = 55$ mm are summarized in Fig. 4b. The three stable states (two operational and

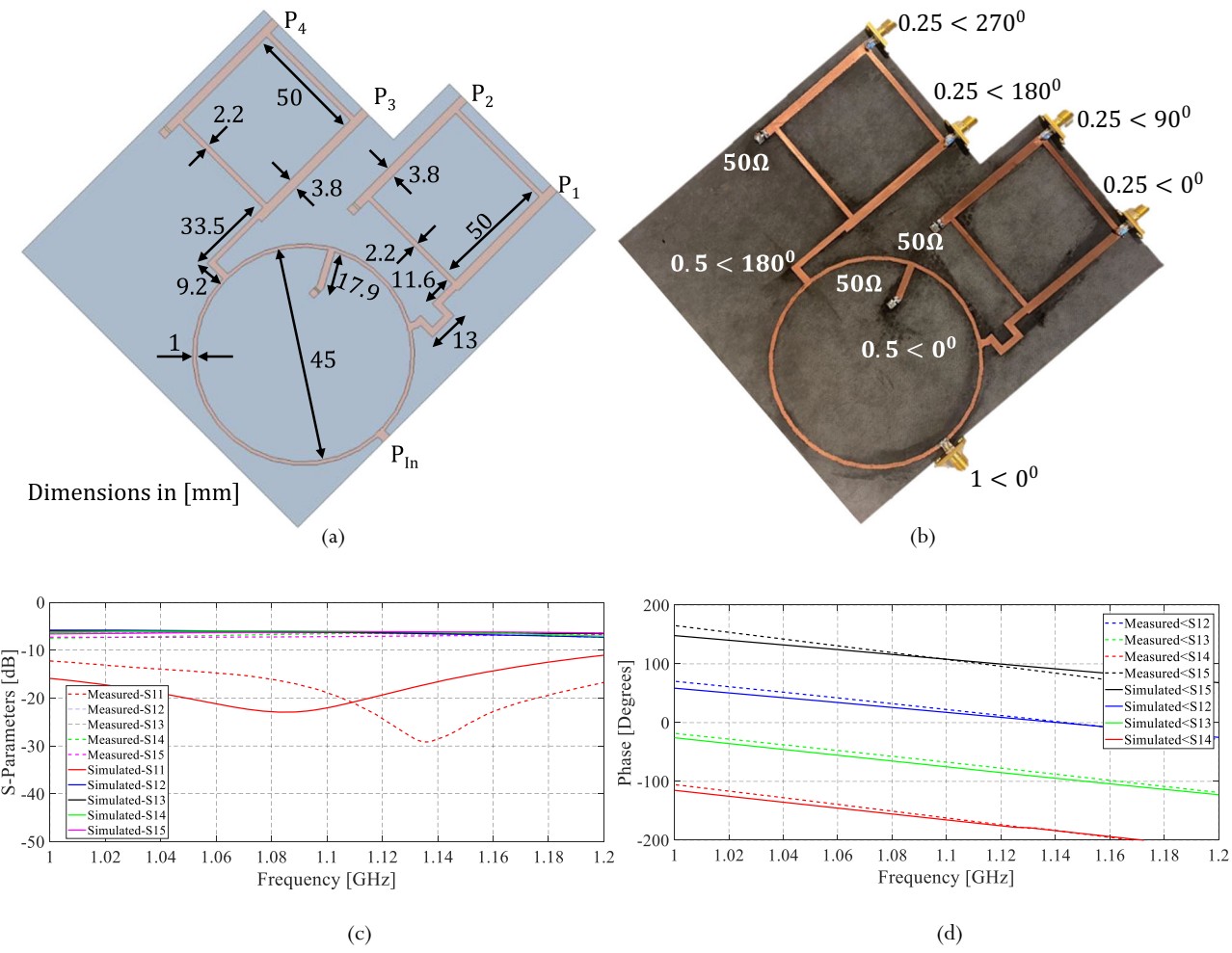

**Fig. 5 | Design of QHA feeding network. a** Proposed feeding network with its dimensions. **b** Fabricated feeding network with amplitude and phase distribution on each port. **c** Simulated and measured S-parameters of the feeding network. **d** The simulated and measured phase difference between different ports.

one non-operational) of the selected design are demonstrated experimentally for a prototype with N = 1.5 turns in Supplementary Video 1.

The use of FRP composites for the multi-stable antenna presents a robust implementation for low-infrastructure environments. Multi-stability of FRP composite structures can be maintained over millions of actuation cycles, despite some stiffness reduction of the material under fatigue[33]. While temperature extremes can shift the location of stable points[34], such effects are expected to be minor over the range of temperatures encountered on Earth and can be mitigated through fiber and polymer selection. These small changes in the stable radius have a modest impact on the antenna gain while overall radiation and polarization characteristics remain stable (Supplementary Table 1). Similarly, small vibrations around the stable states will not have a significant impact on electromagnetic performance (Supplementary Table 1). However, dynamic loading (e.g., wind) that causes resonance can inadvertently switch states of the antenna. In fact, dynamic excitation has been proposed as an active actuation strategy for bi-stable structures[35]. To avoid unintended actuation, the FRP material offers a large design space for tailoring the structural stiffness, and hence resonance frequencies such that they are removed from environmental forcing frequencies. Specifically, we demonstrate that the axial stiffness of the helix can be tuned using the composite fiber angle and strip width while preserving coupling between stable points and antenna reconfiguration (Supplementary Fig. 6). Overall, while state-of-the-art solutions to mechanically reconfigurable antennas trade off

mass and portability with environmental robustness (e.g., heavy motor-driven articulated mechanisms[36] in contrast to membrane-based origami antennas[13]), the technology proposed here offers a lightweight, portable structure with robust multi-stability across a large range of environments.

## The radio frequency (RF) feeding network

The four conductive strips of the QHA are fed by a dedicated feeding network that is fixed underneath the antenna's ground plane. The feeding network, shown in Fig. 5, supplies equal amplitude signals with sequential quadrature phases. As shown in Fig. 5a, the feeding network relies on one rat-race coupler and two 3-dB hybrid couplers[37] (Supplementary Notes 1, 2).

The rat race coupler is a reciprocal four-port network that splits the input signal $P_{in}$ into two equal output signals with a phase shift of 180°. Two of the output ports of the rat race coupler connect to two 3-dB hybrid couplers as shown in Fig. 5. Each 3-dB hybrid coupler is also a four-port network that divides its input signal into two equal output signals ($P_1$ and $P_2$ or $P_3$ and $P_4$, respectively) with a 90° phase shift as shown in Fig. 5b. All remaining ports are isolated and connected to 50Ω loads. The combination of a rat-race coupler and the two 3-dB hybrid couplers leads to a quadrature feeding network with equal amplitude signals and sequentially quadrature phases.

Before testing the functionality of the antenna, the feeding network is assessed. The comparison between the simulated and measured S-parameters magnitude of the proposed feeding network is

shown in Fig. 5c. Figure 5d presents the simulated and measured phase differences between the various ports of the proposed feeding network. These results prove that the designed network provides equal amplitude signals with sequential quadrature phases.

## Ground plane design and sliding feed

The physical realization of the two states of the proposed antenna structure requires sliding the various helical wires along the ground plane as the antenna reconfigures its radius. A customized ground plane is implemented as part of the proposed structure to allow the variation of the antenna's radius. The typical square ground plane is modified to include four slots coinciding with the motion of the four conductive arms as the antenna changes its radius (Fig. 6a).

Four coaxial cables with SMA connectors are connected to the four conductive helical arms and move along these slots. The ground of each SMA connector is attached via conductive epoxy to a slot cover that slides underneath the respective slot. A fixed distance is maintained between the cover and the ground plane using 3D-printed supports. To ensure the electrical connectivity between the ground plane and the sliding covers, spring-loaded pins with a gold-plated ball bearing at their tips are integrated with the 3D-printed parts. Figure 6b shows a partially assembled support for the sliding covers. The coaxial feeding movement on the ground plane with the state change of the antenna is shown in Fig. 6c.

Photos of the top and bottom views of the ground plane are shown in Fig. 7a, b. Photos of the completed assembly can be seen in Fig. 7c. The manufactured ground plane is fabricated using 0.5 mm thick Copper and has a side length of 374 mm. A comparison between the performance of the QHA over a full ground plane with its performance over a covered slotted ground is shown in Supplementary Note 3. It is demonstrated that the effect of the slots is negligible.

## Measured electromagnetic performance of the QHA

The antenna was measured in an anechoic chamber at the Antenna Measurement facility in the EMpact lab at the American University of Beirut. It is a standard ETS Lindgren anechoic chamber[38] connected to a Performance Network Analyser (PNA)[39], a controller, and a computer to control the turning table and extract the results (Methods). The antenna under test is put in transmitter mode and a horn antenna is used as a receiver.

Figure 7d, e shows the antenna with its own feeding network being tested in an anechoic chamber in its two operating states. The measurement was done in the two stable states of the antenna and the performance was compared to simulation.

The transition of the fabricated antenna from State 1 to State 2 and vice-versa is shown in Fig. 7f, where the fabricated antenna in its two states is presented. Transition from State 1 to State 2 is executed as a result of an applied vertical downward force which, when applied to the tips of the QHA, forces the rotational joints to rotate in the clockwise direction. As a result of this rotation, the conductive strips (brown colored strips in Fig. 7f) rotate in a clockwise and downward direction, the non-conductive strips (white colored strips in Fig. 7f) rotate in a counterclockwise and downwards direction. This increases the radius of the QHA, and transitions it into a stable state 2. On the other hand, a vertical upward force applied to the tips of the QHA in stable state 2, forces the rotational joints to rotate in the counterclockwise direction. As a result, the conductive strips rotate in a counter-clockwise and upward direction, the non-conductive strips rotate in a clockwise and upward direction, and the radius of the antenna decreases. As a result, the antenna returns to stable state 1.

The comparison between the simulated and measured input reflection coefficients ($|S_{11}|$ [dB]) is presented in Fig. 8a for both states, demonstrating that the antenna preserves the same operational frequency. The great agreement between measured and simulated $|S_{11}|$ demonstrates that the fabricated antenna prototype along its feeding network satisfies the design requirements in terms of frequency operation.

The radiation characteristics of the fabricated prototype in the two deployed states are similarly assessed and compared to simulations. The radiation patterns of the antenna are measured at $f = 1.1$ GHz

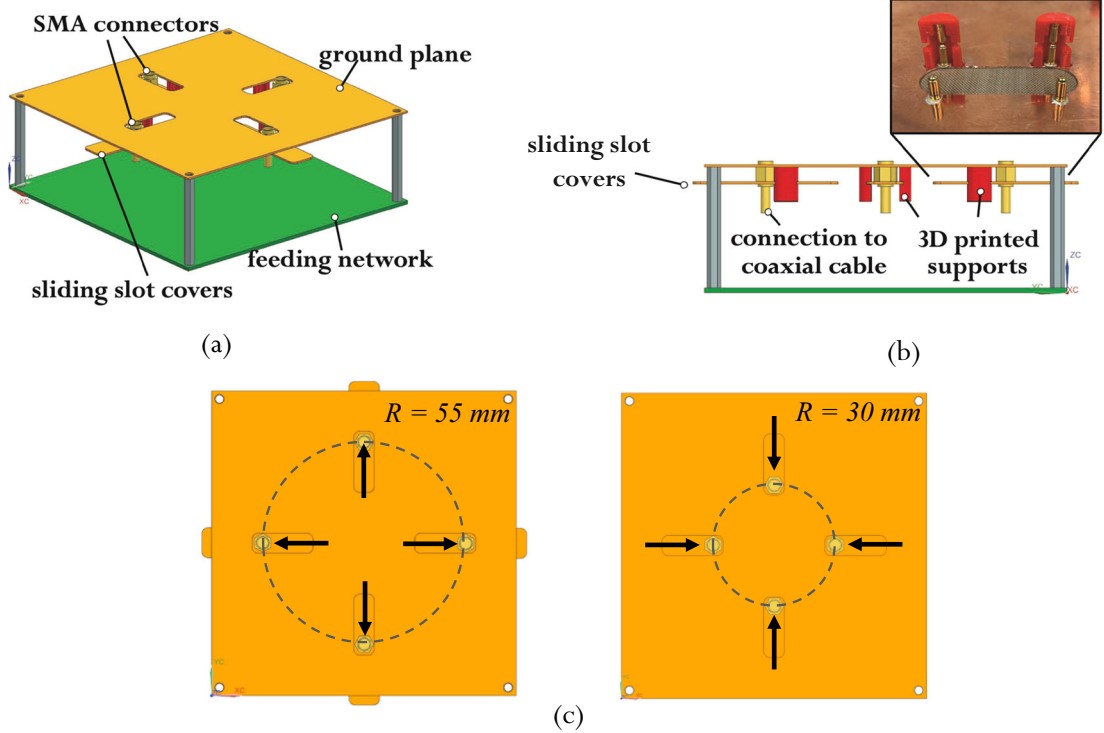

(a)

(b)

(c)

**Fig. 6 | Reconfigurable ground plane. a** Isometric view showing a schematic of the ground plane design. **b** 3D printed support and spring-loaded pins partially assembled with a side view showing a schematic of the ground plane design. **c** Coaxial feeding movement.

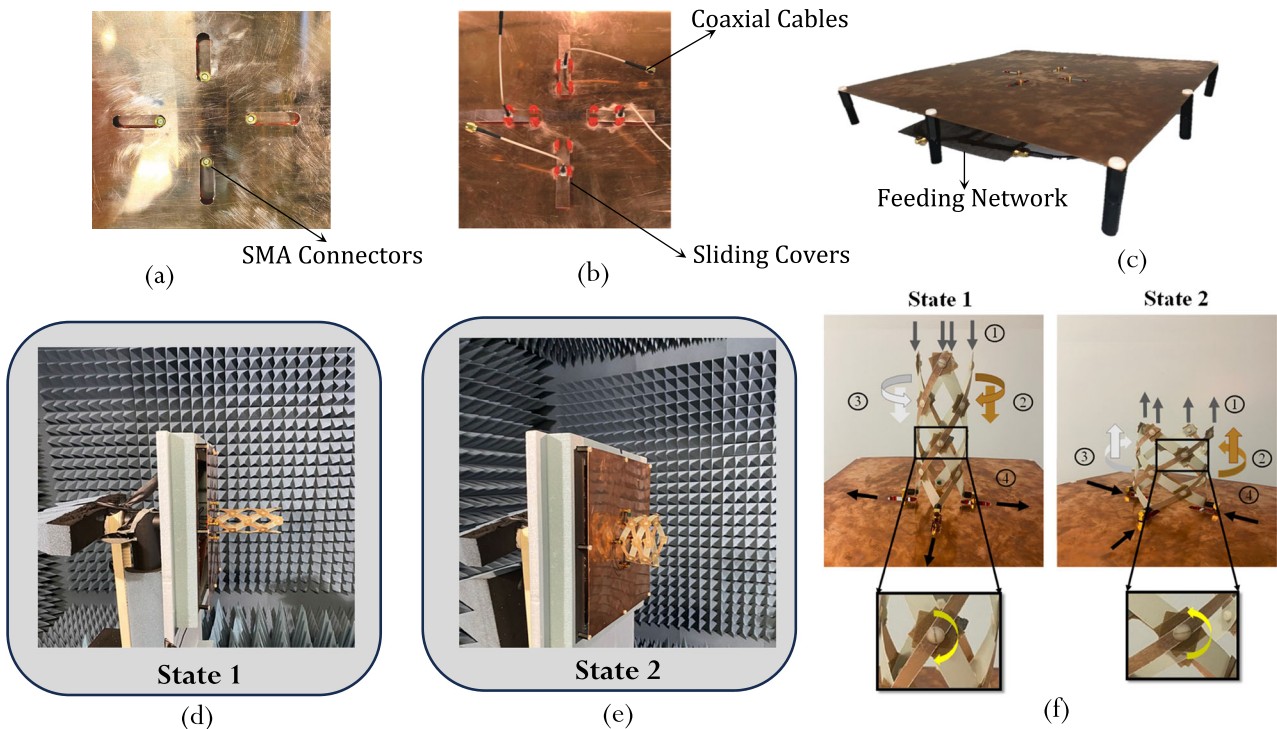

**Fig. 7 | Photographs of physical antenna prototypes. a** Top of the ground plane. **b** Bottom of the ground plane. **c** The full group plane assembly. **d**, **e** Antenna under measurement in the anechoic chamber in states 1 and state 2. **f** Illustration of the structure's transition between state 1 and state 2.

since this is the frequency of interest for the antenna operation. In state 1, the antenna exhibits an almost omnidirectional radiation pattern as shown in Fig. 8b. As for state 2, the antenna exhibits a directional radiation pattern as depicted in Fig. 8c. For both states, the simulated and measured results show good agreement.

The axial ratio of the second state of the proposed antenna structure can be calculated using (3)[40].

$$AR = \frac{2\lambda S}{(\pi D)^2} \quad (3)$$

In state 2, the helix spacing is 0.5λ while its radius is 0.1833λ. These dimensions result in a predicted axial ratio of 0.7546 dB, which is less than the 3 dB threshold. Thus, circularly polarized radiation is obtained in state 2. To validate the circular polarization of the antenna in state 2, the simulated and measured axial ratio at $f$ = 1.1 GHz are plotted in Fig. 8d over the span of elevation angles where the maximum radiation is focused. It is clear that the axial ratio drops to below 3 dB for a range of angles between ± 40° where the maximum radiation is centered, proving that the antenna is circularly polarized.

The antenna exhibits a maximum measured realized gain of 2.6 dB for state 1, while it exhibits a maximum measured realized gain of 8.5 dBic for state 2, which renders state 2 of this antenna suitable for satellite communication. Comparisons between the measured and simulated realized gains for both states of the antenna are shown in Fig. 8e, f. It is clear from Fig. 8 that the measured and simulated results are in excellent agreement, which confirms that the designed antenna achieves its envisioned performance with great fidelity.

**Designed QHA Compared to a Conventional QHA**
To achieve a normal mode conventional helix, the circumference, $C$, of the helix, and the spacing, $S$, between each helical turn, must be less or equal to $\frac{1}{2}$ λ. As a result, for a frequency of 1.1 GHz, the radius, $R$, of the helix must be less or equal to 21.7 mm, and the spacing, $S$, must be less than or equal to 136.0 mm (Supplementary Table 2). On the other

hand, to achieve an axial mode conventional helix[40], $C$ must be between $\frac{3}{4}$ λ and $\frac{4}{3}$ λ in addition to a spacing $S$ of $\frac{1}{4}$ λ. As a result, for a frequency of 1.1 GHz, axial mode operation is achieved when the radius and spacing of the helix[40] are 32.6 mm < R < 58.0 mm, and S ≅ 68.3mm. The conventional QHA calculation studies the helical spacing between the values of 0 and λ . However, in our design we went beyond λ in terms of spacing to venture an approach where we achieve a normal mode and an axial mode corresponding to stable geometries from a mechanical perspective. In other words, an unconventional helical design was selected to merge electromagnetic reconfiguration with mechanical stability.

Hence, a major portion of the analysis executed in this paper is focused on identifying the sweet spot between the mechanical stability of the structure and its electromagnetic performance. To ensure deployment of the QHA while maintaining the same frequency of operation, the radius and spacing must be adjusted to maintain the same strip length. So when the radius of the wire increases, its spacing decreases and vice-versa (as in Eq. (1)).

As can be seen from Supplementary Fig. 5, for a conventional QHA at an operating frequency of 1 GHz the total length of the wire $l$ differs between the normal and axial modes. Therefore, the same conventional antenna cannot operate in two different modes within the same frequency range. Our proposed design overcomes this limitation by preserving the same wire length $l$ for all the radii. This was achieved by modifying the spacing between the wires while taking into consideration the spacing-radius relation (Fig. 2c) ensuring that the mechanical stability of the design is preserved.

We can generalize the design space of reconfigurable electromagnetic performance by representing it as a function of the operating wavelength, λ (Fig. 9a). In this way, the discovered reconfiguration principle can be applied to any desired frequency. As can be seen, for a QHA with 0.5 turns, 0.935λ < S < 1.14 λ, and 0.0833λ < R < 0.133λ, produces an omnidirectional-like pattern. A QHA with 0.5 λ < S < 0.935 λ and 0.133 λ < R < 0.18833 λ is circularly polarized (CP) with a directional pattern. Beyond these limits, the antenna is not resonant (NR). A

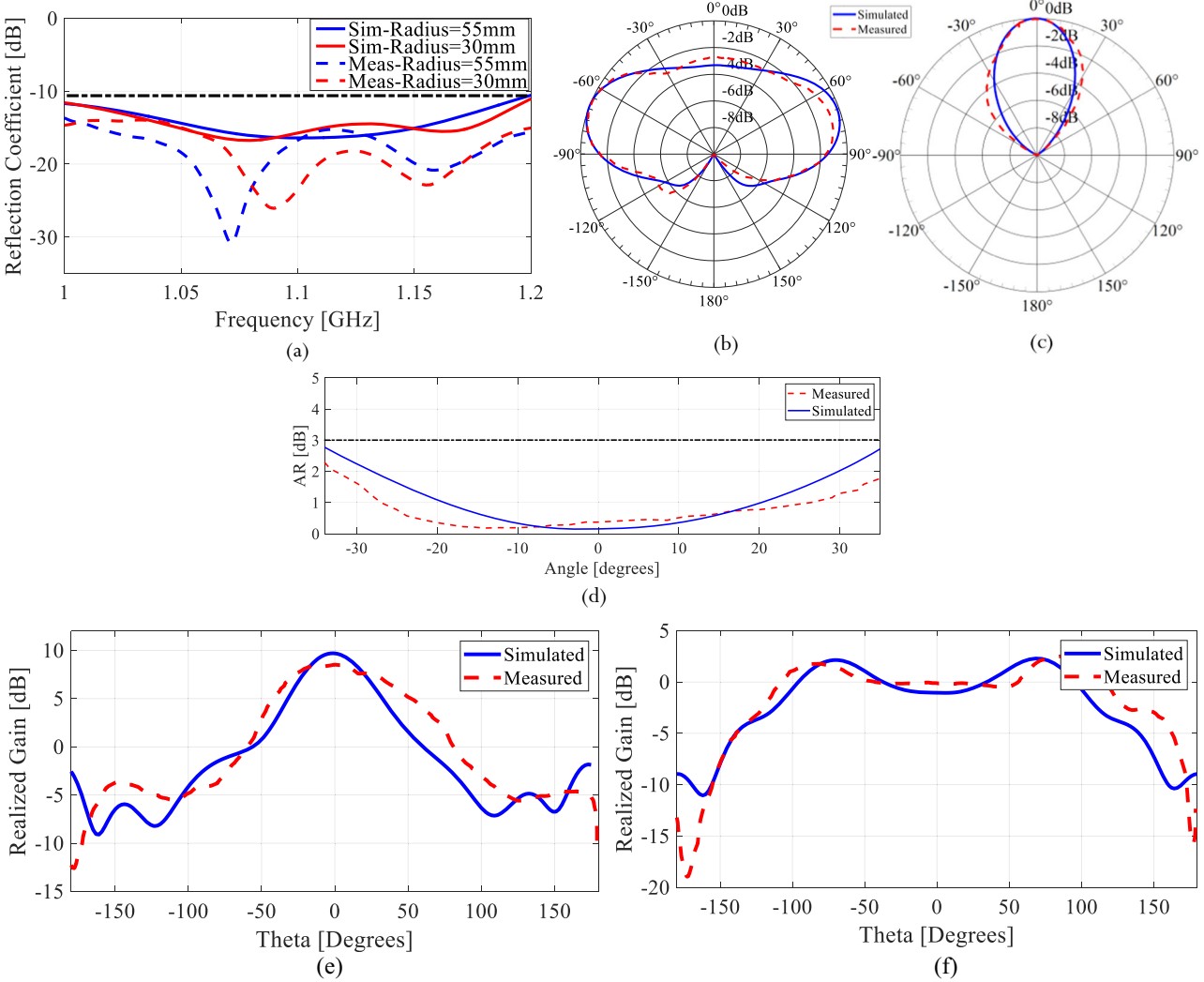

**Fig. 8 | Comparison between measured and predicted antenna performance in the two operating states at 1.1 GHz. a** Reflection coefficient. **b** The normalized measured and simulated radiation pattern for State 1. **c** The normalized measured and simulated radiation pattern for State 2. **d** Measured and simulated Axial Ratio in dB for State 2. **e** Comparison between measured and simulated realized gain at the operational frequency and for the azimuthal angle (phi = 0) plane cut for variable elevation angle (theta) for state 2 and **f** state 1.

demonstration of the radiation pattern reconfiguration across a range of operating frequencies is provided in Fig. 9b.

### Comparing passive and active actuation

Multi-stability offers a user-centric approach to antenna actuation that eliminates the need for DC power. This means users can manually adjust the antenna to meet their communication requirements, knowing that multi-stability will precisely configure the structure for optimal electromagnetic performance. This operational principle proves especially advantageous in regions with limited infrastructure and power resources.

Alternatively, a multi-stable antenna can also be autonomously reconfigured via powered actuators. It is noteworthy that in this case, multi-stability still offers significant energy savings as the actuation needs to be powered only to reconfigure the structure while multi-stability maintains the antenna geometry in the desired operational states. In addition, the combination of multi-stability and powered actuation can ensure that the antenna maintains its current state in case of power failure and may be advantageous where safety considerations preclude manual reconfiguration. Stimuli-responsive smart materials are an attractive option for the actuation of multi-stable structures[35,41,42] instead of traditional electro-mechanical actuators

such as motors. Smart material actuators can interface directly with the structure[41] or be embedded in the material[35,42] to reduce the mass associated with power transmission (e.g., gears). For the antenna proposed here, shape memory alloys present one actuation possibility, if the antenna is proposed for implementation in different circumstances or environments where power is available and not scarce. These materials exhibit phase transitions[43] that allow the actuator to hold a large deformation that can be recovered via resistive heating. A pair of shape memory alloy actuators can be used in an antagonistic arrangement for reversible actuation[41,44]. We present a proof-of-concept antenna deployment using shape memory alloy spring actuators in Supplementary Fig. 7, however, we emphasize the fact that such actuation is only useful in regions where power resources are available contrary to our intended implementation in this work.

On the other hand, the deployment mechanism adopted for this structure eliminates the need to integrate PIN diodes or any other electronic switch into the feeding network or the antenna itself. Our proposed passive bistable deployment technique offers more robustness, and a greater immunity to environmental factors, and eliminates any nonlinearity effects that may result from the integrated switches. Furthermore, a bias-free system implicates less interference in the antenna's radiation characteristics, especially in a QHA topology

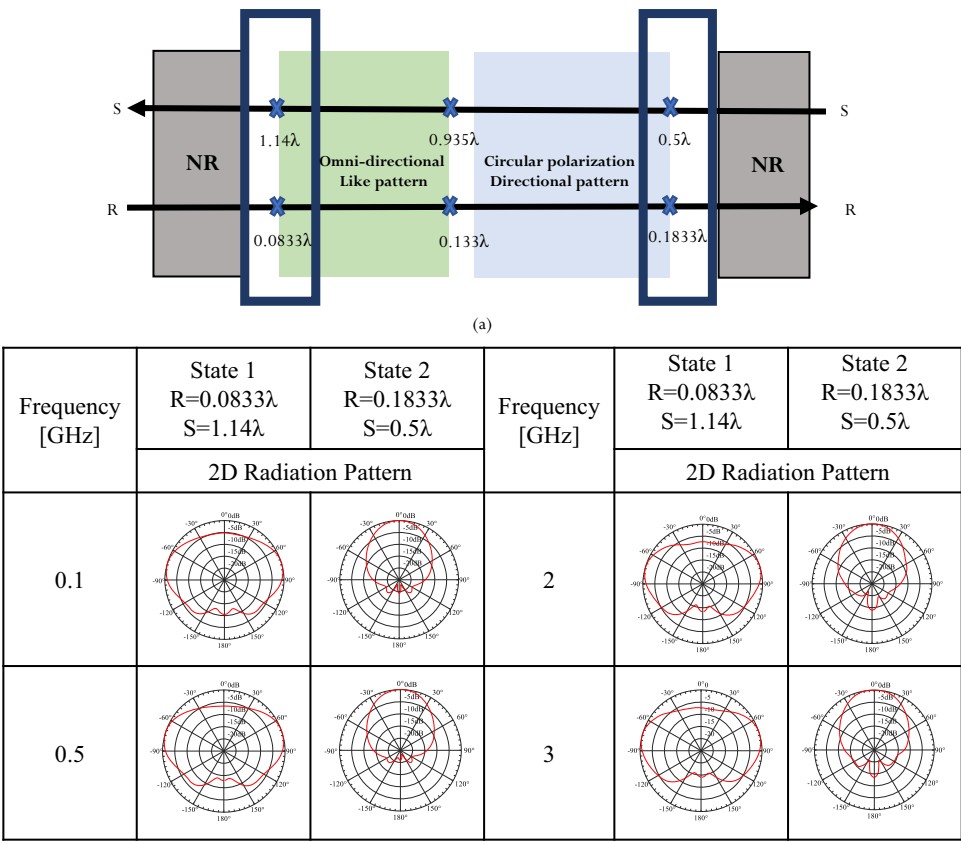

(a)

| Frequency [GHz] | State 1<br>R=0.0833λ<br>S=1.14λ | State 2<br>R=0.1833λ<br>S=0.5λ | Frequency [GHz] | State 1<br>R=0.0833λ<br>S=1.14λ | State 2<br>R=0.1833λ<br>S=0.5λ |
|---|---|---|---|---|---|
| | 2D Radiation Pattern | | | 2D Radiation Pattern | |
| 0.1 | | | 2 | | |
| 0.5 | | | 3 | | |

(b)

**Fig. 9 | Scaling principles for antenna reconfiguration. a** The design space for antenna performance reconfiguration as a function of geometry. **b** Radiation pattern reconfiguration across a wide range of operating frequencies.

which is an out-of-plane topology. Additionally, it is important to emphasize that realizing the unique reconfigurable functionality of our proposed QHA necessitates a mechanical modification of its structural topology, a feature that cannot be achieved using electronic switching reconfiguration.

## Discussion

This work presented a mechanically and electromagnetically reconfigurable QHA for low-infrastructure areas. The work revealed regions of the design space beyond the traditionally used geometries for helical antennas, which enabled large reconfiguration of the antenna's radiation and polarization characteristics. Furthermore, the anisotropy of lightweight FRP composite materials was leveraged to create stable self-locking geometries corresponding to the different electromagnetic operating regimes, allowing them to maintain multiple geometries passively.

Specifically, we have demonstrated an enhanced antenna functionality allowing both a high gain, directional, and circularly polarized operation for satellite communication and a nearly omnidirectional operation for device-to-device ground communication. Meanwhile, the operating frequency of the antenna in the L-band was kept fixed.

The novelty of this work lies in coupling electromagnetic and mechanical reconfiguration throughout the design process. This is further highlighted by the proposed design guidelines enabling scaling of the identified principle beyond a specific design. While previous work has considered the use of mechanical bi-stability for antennas, bi-stability was decoupled from the antenna operation. Deployable antennas where the stowed and deployed states are both mechanically stable are presented in literature[45,46]. In this case, only the deployed state operates as an antenna. Other studies have investigated the effects of particular bi-stable structures on electromagnetic

performance[47,48]. These studies showed that while some minor reconfiguration of operating frequency or beam direction was possible, the antenna performance, in general, was significantly limited by the constraining structures of bi-stable geometries. For example, bi-stable composite laminates were employed in the design of a microstrip patch antenna[49], or as part of composite tape springs for deployable monopoles[45] out of a CubeSat shroud. On the other hand, our presented design herein is advantageous over available work in the literature as we demonstrated a truly coupled performance between mechanical stability and electromagnetic performance, without resorting to any form of external actuation. On the other hand, existing passive and deployable reconfiguration of multifunctional antennas presented in the literature[50,51] do not exhibit the same capability in antenna performance as they focus on single parameter reconfiguration, typically the frequency of operation.

The impact of this approach is evident by examining implications for communications in low-infrastructure areas. We presented a lightweight, portable, and low-energy solution for low-infrastructure areas with the required agile antenna performance. The use of lightweight FRP composites for the antenna enables significant weight savings (the antenna with a total weight of 39 g and the ground plane with a weight of 737 g) compared to metallic dishes (~20 kg) typically deployed in disaster-stuck areas[2]. Additionally, the bi-stability feature removes the requirement for continuous DC power that is needed for mechanical reconfiguration. In fact, it is important to note that, in this work, we have intentionally avoided the use of active components for antenna reconfiguration. Active components such as switches or actuators are abundant in the literature for reconfiguring antenna structures of various topologies and serving different implementations such as in QHAs[52,53]. However, antennas that depend on electrical reconfiguration require additional biasing power, which must be

accounted for as part of the power budget of the design, a constraint that we are minimizing in our design.

The main objective of our work is to propose a solution that can restore communication in disaster-stricken regions where a shortage or lack of power is a reality. Hence, by proposing a passive reconfigurable bi-stable antenna structure we ensure antenna agility, which enables high-performance operation in multiple communication modes. In other words, coupled mechanical and electromagnetic design has enabled multi-functionality in impromptu communications.

The proposed antenna structure is envisioned to be practically operational by connecting it with an off-the-shelf RF transceiver[54] that can be integrated easily with the antenna structure. An RF transceiver that is battery-powered through laptop integration[54] can be used to enable the transmit/receive modes through the antenna as its front-end element. Once this transceiver is connected to the antenna, its required input power will be the only needed power to operate the system. Moreover, our antenna in its two states of operation ensures that communications is achieved, whether in state 1 for terrestrial communication with an almost omnidirectional pattern or in state 2 where its radiation is directive with a realized gain of 8.5 dBic. When comparing the proposed antenna structure with existing antennas that have been proposed for satellite communication as discussed in[55], our proposed structure can maintain the communication capability from its polarization and gain requirements while providing an additional passive reconfiguration mechanism.

## Methods

### Prediction of stability points

An analytic energy-based model that computes the 2-dimensional deformation of the helical strips during the deformation of the lattice is employed to design the FRP composite material that the helical strips are made from[29,56]. The aim of the model is to design a structure with stable geometries corresponding to the two desired operating states of the antenna. In the most general case, the strain energy of a linear elastic composite material strip due to in-plane and bending deformation is given by:

$$U_{strip} = \frac{1}{2} \iint_A \begin{bmatrix} \boldsymbol{\varepsilon}^0 \\ \boldsymbol{\kappa} \end{bmatrix}^T \begin{bmatrix} \boldsymbol{A} & \boldsymbol{B} \\ \boldsymbol{B} & \boldsymbol{D} \end{bmatrix} \begin{bmatrix} \boldsymbol{\varepsilon}^0 \\ \boldsymbol{\kappa} \end{bmatrix} du\,dv \qquad (4)$$

where the integral is over the area of the strip and u and v are local coordinates along the length and width of the strip, respectively. **A, B**, and **D** are 3×3 matrices representing the FRP composite stiffness relating the force, **N**, and moment, **M**, resultants per unit width to the mid-plane strains, $\boldsymbol{\varepsilon}^0$, and curvatures, $\boldsymbol{\kappa}$[57],

$$\begin{bmatrix} \boldsymbol{N} \\ \boldsymbol{M} \end{bmatrix} = \begin{bmatrix} \boldsymbol{A} & \boldsymbol{B} \\ \boldsymbol{B} & \boldsymbol{D} \end{bmatrix} \begin{bmatrix} \boldsymbol{\varepsilon}^0 \\ \boldsymbol{\kappa} \end{bmatrix} \qquad (5)$$

Note that **ABD** stiffness is distinct for the conductive and dielectric strips. We use a 2D model following work in the literature[29,58] for the strip curvature assuming that there is longitudinal and transverse bending as well as twisting in the strips. Only longitudinal strains are considered, with transverse and shear strains assumed to be negligible. From this, we deduce that the largest influence on the stability is the material bending stiffness, **D**, and the strip width (which has a large effect on the transverse curvature of the strips).

The strip energy in Eq. (4) is summed for all the helical strips in the antenna and the stable points correspond to the minima of the energy. The axial forces exerted during deformation can be computed using Eq. (2).

### Selected antenna design

The large mechanical design space is narrowed down using two constraints. First, only symmetric laminates were considered to prevent

**Table 1 | Summary of structural and material parameters of quadrifilar helix antenna**

| | |
|---|---|
| Conductor width [mm] | 8.0 |
| Dielectric width [mm] | 14.5 |
| Strip length [mm] | 188.5 |
| Conductive Strip Layup [deg] | $[8_7^{GFRP}/0_{pw}^{PB}/8_7^{GFRP}]$ |
| Dielectric Strip Layup [deg] | $[-8_4^{GFRP}/90_3^{GFRP}]_s$ |

the helical strips from warping in response to a temperature change during manufacturing. Second, equal strip lengths (defined along the arc length of the helix) for all strips are selected to prevent torsion in the helix. The strip length is fixed at $l = 188.5$ mm to allow operation at approximately 1 GHz. These constraints leave the following variables for the design: the widths of the dielectric and conductive strips (defined transverse to the length of the strip), and the layups of the conductive and dielectric strips (i.e. number of layers in the composite material and the fiber orientation in each layer). The authors previously demonstrated that fiber angles around 10° result in tri-stable lattices with stable states at the minimum and maximum radii as well as an additional state at an intermediate helix configuration[56]. This effect is shown here for the antenna structure in Fig. 4a. These fiber angles result in the highest ratio of bend-twist coupling vs. torsional stiffness for a unidirectional composite (i.e., $D_{16}/D_{66}$). Starting from this design, layers with fibers either parallel or perpendicular to the length of the strips were added at the center of the composite to improve transverse strength. The final structural parameters of the antenna design are summarized in Table 1.

**Materials and manufacturing.** All helical strips are manufactured from E-glass fiber-reinforced epoxy composites. The conductive strips contain a thin woven phosphor-bronze mesh (292 g/m²) at the mid-plane of the laminate. Glass fiber-epoxy prepreg with an areal weight of 25 g/m² is used and is laminated to the desired layup, and it is cured for 2 hours at 120 °C in an autoclave. The material properties of a single glass fiber-epoxy layer are summarized in Supplementary Table 3. The cured material is cut to size to realize the strips. A stencil is used for drilling holes at the intersection points of the strips and these are assembled into a helix using nylon screws. To reduce joint friction, Teflon washers are used between the strips. Friction in the joints can affect the stable points and reduce repeatability.

**Simulation of the antenna performance.** Each of the two stable states of the antenna was simulated using Ansys HFSS[30] to evaluate the electromagnetic performance of the antenna structure. The non-conductive strips are modeled as a dielectric material with a relative permittivity of 3.7 and a loss tangent of 0.001. The number of turns of each helical wire is equal to 0.5 while its ground plane is 374 × 374 mm².

**Measurement of the radiation pattern, polarization and gain.** Testing the antenna is carried out in a standard anechoic chamber from ETS Lindgren[38] at the American University of Beirut. The Anechoic cells operating in the radio frequency (RF) waveband must be insulated and isolated from electromagnetic noise coming from outside the test field[59]. The anechoic chamber contains a space that is lined with radio wave-absorbing material. These absorbers are usually in the form of pyramid-shaped wedges that cover the inside surfaces of the chamber. The test cell structure is designed to resemble a Faraday cage, with a metal sheet serving as the external skin. The absorbers absorb the reflections of the reference signal. At one end of the chamber a horn radiator is placed operating at microwave frequencies. The antenna under test (AUT) is mounted on a rotational table at the other end of the chamber. Elevation and azimuth plots can be plotted by positioning the antenna in the correct plane for each test[38]. The chamber

used for testing the antenna is a fully anechoic chamber, where all surfaces comprising the floor are non-reflective (energy absorbent).

## Reporting summary
Further information on research design is available in the Nature Portfolio Reporting Summary linked to this article.

## Data availability
All data supporting the findings of this study are available within the paper and its supplementary information files. Source data are provided with this paper.

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

## Acknowledgements

This work was supported by the 2019 Innovation Starting Grant (2019 – project 03, awarded to M.S. and J.C.) from the Swiss State Secretariat for Education, Research, and Innovation administered by the University of Applied Sciences and Arts of Western Switzerland (HES-SO). The authors would also like to acknowledge the help of Dr. Fatima Asadallah from the American University of Beirut during the measurement of the antenna prototype.

## Author contributions

J.C., Y.T., and M.S. conceived the concept and design. R. B. designed and simulated the quadrifilar helix antenna and its ground plane. M.S. designed, simulated, and fabricated the antenna and its ground plane. R.B. designed and simulated the antenna feeding network. J.C., Y.T., and R.B. measured the antenna performance and the performance of the feeding network. All authors reviewed the results, provided essential reviews of the manuscript, and approved the final version of the paper.

## Competing interests

The authors declare no competing interests.
