## [Peer review file · Nature Communications]

REVIEWER COMMENTS

Reviewer #1 (Remarks to the Author):

The authors have proposed and studied a manually deployable Quadrifilar Helix Antenna (QHA) with radiation pattern/polarisation reconfigurability. The performance of two states of QHA was simulated and tested in experiment.

The proposed reconfigurable mechanism of the QHA seems interesting. However, the basic idea of this work was already presented in a major conference in this area [28]. For the application mentioned by the authors, a manually deployable QHA is not convenient. A few reconfigurable QHAs through electronical means were reported recently [a, b]. I would suggest the authors to add a mechanical actuation set-up to reconfigure the QHA through an electrical means and re-submit their work to an IEEE journal.

a. X. Yi, et al, 'Polarization and Pattern Reconfigurable Cuboid Quadrifilar Helical Antenna', IEEE TRANSACTIONS ON ANTENNAS AND PROPAGATION, VOL. 66, NO. 6, JUNE 2018.

b. W. Sun, et al, 'A Pattern Reconfigurable Circularly Polarized Quadrifilar Helix Antenna Through Phase Control', IEEE TRANSACTIONS ON ANTENNAS AND PROPAGATION, VOL. 70, NO. 9, SEPTEMBER 2022.

Reviewer #2 (Remarks to the Author):

The authors present a novel mechanically and electromagnetic reconfigurable quadrifilar helix antenna. The use of FRP composites allowed a self-stable locking geometry that corresponded to a different electromagnetic operating regimes.

The supplementary video demonstrated the disconnected antenna's ability to maintain its differing operational regimes based on minor inputs and switched fully with a large enough force applied.

There is a need for fieldable ruggedized antennas for harsh and remote environments, and the main structure of the antenna fits that need, but there are a few questions that would need to be answered, such as:

1. what is the expected lifetime or has lifetime analysis of the operational locking geometry being performed,
2. what are temperature extremes effect on both lifetime of the joints and self-stability of the joints performance (such as, does being in the dessert or artic cause the joints to bind),
3. vibration analysis of structure and its effect on electromagnetic performance,
4. if this structure is placed outdoors in high winds could some mechanical resonance be achieved that causes it to flip between the bi-stable states (self-resonance of a structure causing failure- similar to the Tacoma Narrows bridge collapse),
5. how does this technology compares in performance (lifetime, antenna applications, environment ranges, etc.) to other mechanical reconfigurable antenna technologies,
6. and lastly, does this technology have any other applications or examples of being applied to other antenna types?

Reviewer #3 (Remarks to the Author):

This article proposes a Deployable Quantum-Hall Array (QHA) that is capable of passively

reconfiguring its polarisation state and radiated beam direction. The study exhibits a notable level of interest; yet, there are several comments that necessitate attention from the authors.

1) Why did the author choose for passive reconfiguration instead? In today's world, a significant proportion of systems and processes operate in an autonomous or active way.

2) Line 91- Is it correct the number of turns of the helix is only 0.5?

3) Line 259 - "the spacing is greater than lambda to venture an approach where we achieve a normal mode and an axial mode corresponding to stable geometries from a mechanical perspective". What is the rationale for the necessity of increasing the spacing, considering that the established standard range has already demonstrated its capability to attain both normal mode and axial mode? If the spacing and radius are increased, it is expected that the size of the antenna will increase as well.

4) Figure 2: In practical scenarios, how can the transition from State 1 to State 2 be implemented?

5) Line 146 and Line 299 - Is there any empirical evidence or analysis available to support the claim that the proposed design exhibits low energy consumption?

6) Line 101 - 104 - Are there any figures available to support or substantiate those statements?

We sincerely thank the reviewers for their constructive feedback, which helped us improve the technical quality of the paper. We have carefully revised the paper to address the reviewers' comments. In this document, the comments are presented in blue followed by our point-by-point responses.

Reviewer 1 Comments:

R1C1: The authors have proposed and studied a manually deployable Quadrifilar Helix Antenna (QHA) with radiation pattern/polarization reconfigurability. The performance of two states of QHA was simulated and tested in experiment. The proposed reconfigurable mechanism of the QHA seems interesting. However, the basic idea of this work was already presented in a major conference in this area [28].

Response to R1C1: We thank the reviewer for the comment, which allows us to compare the work presented in this manuscript with the work presented in [28]. In fact, [28] is a brief conference paper of two pages in length. It presents the basic idea of the proposed deployable antenna with only simulated results. On the other hand, in this manuscript, we present the entirety of the work in both simulation and measurement and from both structural and electromagnetic perspectives. In this work, we present the fully implemented and integrated feeding network and reconfigurable ground plane as well as the testing of the entire system in all its states of operation. The manuscript is also enriched with a deep dive analysis into the electromagnetic and mechanical performance of the proposed design with plenty of details on the fabricated prototype. Moreover, we present in this manuscript a comparison between the conventional QHA and our proposed structure (Lines 283-316). Furthermore, a generalized equation for the design of a reconfigurable QHA at any frequency was formulated with the prediction methods of stability points discussed.

R1C2: For the application mentioned by the authors, a manually deployable QHA is not convenient. A few reconfigurable QHAs through electronical means were reported recently [a, b]. I would suggest the authors to add a mechanical actuation set-up to reconfigure the QHA through an electrical means and re-submit their work to an IEEE journal

a. X. Yi, et al, 'Polarization and Pattern Reconfigurable Cuboid Quadrifilar Helical Antenna', IEEE TRANSACTIONS ON ANTENNAS AND PROPAGATION, VOL. 66, NO. 6, JUNE 2018.

b. W. Sun, et al, 'A Pattern Reconfigurable Circularly Polarized Quadrifilar Helix Antenna Through Phase Control', IEEE TRANSACTIONS ON ANTENNAS AND PROPAGATION, VOL. 70, NO. 9, SEPTEMBER 2022

Response to R1C2: We agree with the reviewer that electrically reconfigurable QHAs are proposed in the literature such as the ones proposed by the reviewer [a,b]. However, the fact that these antennas require electrical reconfiguration necessitates additional power requirements to be imposed on the structure. A constraint that we are trying to minimize in our design. Hence, we have intentionally avoided the use of active components such as switches or actuation mechanisms, which would require external biasing and hence, additional power. The objective of our proposed work is to restore communication in disaster-stricken regions where a shortage/lack of power presents a major issue, hence we resorted to designing a manually deployable antenna. The proposed deployable antenna does not need any additional power (dc power, etc.) to reconfigure its electromagnetic characteristics. The manual deployable and passive reconfiguration is possible

due to the bi-stability characteristic of the structure, which not only enables state switching but also maintains the stability of the structure once a state is reached.

We have rewritten parts of the section “Discussion” (lines 370-380) in the revised manuscript to address this issue as well as we added references (a) and (b) as references (53), and (54) in our revised manuscript.

On the other hand, we do agree with the reviewer that autonomous mechanical actuation may be advantageous in other environments for convenience. As such, we propose the use of shape-memory alloy actuators for reconfiguration in such scenarios. These are metallic alloys that demonstrate a temperature-dependent transformation in their crystal structure. They can be stretched to retain strains of ~5% axially that can be recovered upon heating due to the shape memory effect. Two shape memory actuators can be used antagonistically to achieve reversible actuation. While the demonstration of reversible actuation is beyond the scope of the work, we added a proof-of-concept of antenna deployment using shape memory alloy springs in Supplementary Figure 7 and it is shown below.

Supplementary Figure 7: Deployment demonstration using shape memory alloy springs. Shape memory alloy springs are stretched to the desired actuation stroke and resistive heating is used to heat up springs allowing them to recover the stroke (shape memory effect) and deploy the antenna. The springs are made of NiTi with an austenite finish temperature of 45°C, a wire diameter of 1 mm, and a spring diameter of 4.75 mm. A fixed actuation current of 5A is used. The unstretched and stretched lengths of the actuators correspond to stable states 1 and 2, respectively.

We added a section ‘Comparing Passive and Active Actuation’ to the main text to summarize this discussion (Lines 318-336) as well as supporting references (35), (42-45).

We would like to thank Reviewer 1 again for taking the time to review the paper and provide feedback that helped us further improve the content of the paper and we hope that we have answered all the concerns that the reviewer presents.

Reviewer 2 Comments:

The authors present a novel mechanically and electromagnetic reconfigurable quadrifilar helix antenna. The use of FRP composites allowed a self-stable locking geometry that corresponded to a different electromagnetic operating regime. The supplementary video demonstrated the disconnected antenna's ability to maintain its differing operational regimes based on minor inputs

and switched fully with a large enough force applied. There is a need for fieldable ruggedized antennas for harsh and remote environments, and the main structure of the antenna fits that need, but there are a few questions that would need to be answered, such as:

R2C1. what is the expected lifetime or has lifetime analysis of the operational locking geometry being performed,

Response to R2C1: Research has demonstrated that bi-stable structures made of fiber-reinforced polymer (FRP) composites continue to exhibit bi-stability after millions of fatigue cycles, despite a small reduction in material stiffness due to fatigue. **We added a comment regarding these results in lines 165-167 and have added reference (33) to the revised manuscript.**

R2C2. what are temperature extremes effect on both lifetime of the joints and self-stability of the joints performance (such as, does being in the dessert or artic cause the joints to bind),

Response to R2C2: The stability of composite structures can be affected by temperature extremes, as happened on the composite radar instrument on the Mars Express spacecraft orbiting Mars when an attempted deployment of the radar at -70°C showed an additional stable state not observed at room temperature. In this work, we anticipate operation in more moderate temperature ranges encountered on Earth. The stable states are expected to shift only slightly due to the temperature-dependent moduli of the composite material constituents. This can also be minimized through careful material selection. Small shifts in the stable states are expected to have a small impact on the gain while the radiation pattern characteristics remain stable (this is demonstrated in Supplementary Table 1). Regarding the joints, the most important design consideration is to reduce friction in the joints. In this work, we use Teflon spacers between the strips to achieve this. A more robust implementation can use sealed bearings and an appropriate lubricant to provide reliable operation in dusty environments across a broad range of temperatures. These components are available off-the-shelf.

We summarize this discussion by adding lines 167-171 to the main text of the revised manuscript as well as reference (34).

R2C3. vibration analysis of structure and its effect on electromagnetic performance,

Response to R2C3: Similar to how small changes in the stable radius do not impact the overall radiation pattern significantly, small vibrations around the stable states are also expected to preserve the radiation pattern with a small impact on the gain (**Supplementary Table 1**). **We have added lines 171-172 to the main text of the revised manuscript to summarize this.**

R2C4. if this structure is placed outdoors in high winds could some mechanical resonance be achieved that causes it to flip between the bi-stable states (self-resonance of a structure causing failure- similar to the Tacoma Narrows bridge collapse),

Response to R2C4: Larger vibrations can cause resonance and switching between stable states. This can be used as an active actuation strategy or avoided by shifting the structure's resonance frequencies in both stable states away from environmental forcing frequencies (e.g., wind). **We**

added an analysis to demonstrate that the axial stiffness of the structure in the two stable states (and hence the resonance frequencies) can be tuned without impacting the location of the stable states. We have also added Supplementary Figure 6 to summarize this analysis.

Supplementary Figure 6. Tailoring axial stiffness of two stable states, K_1 and K_2 , using the fiber angle, θ , and dielectric strip width, w . The following are used for the analysis: fixed conductive strip width of 8 mm, conductive strip layup of $[\theta_7^{GFRP}/0_{pw}^{PB}/\theta_7^{GFRP}]$, and dielectric strip layup of $[-\theta_4^{GFRP}/90_3^{GFRP}]_s$. (a) Stable radius in state 1. Note that the stable radius of state 2 is always at the maximum possible radius. (b) Resulting axial stiffness in two stable states. The axial stiffness can be tailored to control the structural resonance in the two states under the constraint of a desired stable radius (e.g., $R = 30$ mm as in the red line in (a)).

In addition, we added a few lines summarizing this discussion and new analysis to the main text in lines 173-179.

R2C5. how does this technology compares in performance (lifetime, antenna applications, environment ranges, etc.) to other mechanical reconfigurable antenna technologies,

Response to R2C5: The greatest advantage of this technology over other mechanically reconfigurable antennas is that we achieve large electromagnetic reconfiguration while simultaneously achieving stable operating states, thereby reducing the energy consumption of the system. This is achieved by coupling the mechanical and electromagnetic design. **Advantages over other bi-stable designs are discussed in detail in lines 350-363.**

Additionally, the proposed technology presents a lightweight design leveraging compliant mechanisms for reconfiguration while the FRP composite material provides robust operation in a range of environments. This is in contrast to existing concepts that are either portable or environmentally robust. **We added few lines to the main text in lines 179-182 to summarize this discussion.**

R2C6. and lastly, does this technology have any other applications or examples of being applied to other antenna types?

Response to R2C6: The use of bistable material has been discussed in the literature for applications in deployable antennas or deployable booms for various space structures. However, while previous work has considered the use of mechanical bi-stability for antennas, bi-stability

was decoupled from the antenna operation. Deployable antennas where the stowed and deployed states are both stable are presented in the literature as in references (46) and (47) in the revised manuscript. In this case, only the deployed state operates as an antenna. Other studies have investigated the effects of bi-stable structures on electromagnetic performance as discussed in references (48) and (49) of the revised manuscript. While some minor reconfiguration of operating frequency or beam direction was demonstrated, the antenna performance was significantly constrained as the bi-stable geometries could not be modified. For example, bistable composite laminates were employed in the design of a microstrip patch antenna as presented in reference (50), or as part of composite tape springs for deployable monopoles out of CubeSat shroud as in reference (46). However, our presented design can be considered more advantageous than the available work in the literature as we demonstrated a truly coupled performance between mechanical stability and electromagnetic performance, without resorting to any form of external actuation. On the other hand, existing passive and deployable reconfiguration of multifunctional antennas presented in the literature as in references (51) and (52) do not exhibit the same capability in antenna performance as they focus on a single parameter reconfiguration typically the frequency of operation without additional controllable reconfigurable performance metrics.

The third paragraph of the discussion section was modified to reflect this discussion (lines 350-363).

We would like to thank Reviewer 2 again for taking the time to review the paper and provide feedback that helped us further improve the content of the paper and we hope that we have answered all the concerns that the reviewer presents.

Reviewer 3 Comments:

R3C1: This article proposes a Deployable Quantum-Hall Array (QHA) that is capable of passively reconfiguring its polarisation state and radiated beam direction. The study exhibits a notable level of interest; yet, there are several comments that necessitate attention from the authors. Why did the author choose for passive reconfiguration instead? In today's world, a significant proportion of systems and processes operate in an autonomous or active way.

Response to R3C1: The antenna is purposefully designed to operate in disaster-stricken regions where power is scarce if not lacking. As a result, passive reconfiguration achieves this goal by providing a dynamic antenna operation without spending additional non-existing power on the actuation mechanism. It enables the antenna to achieve dual functionality without requiring additional power. **The tradeoff between passive and active actuation is discussed in a newly added section in lines 318-336.**

R3C2: Line 91- Is it correct the number of turns of the helix is only 0.5?

Response to R3C2: Yes, indeed, the number of turns of the helix antenna is only 0.5. The provided supplementary video demonstrates the bi-stability and deployment of the structure on a longer design.

R3C3: Line 259 - "the spacing is greater than lambda to venture an approach where we achieve a normal mode and an axial mode corresponding to stable geometries from a mechanical

perspective". What is the rationale for the necessity of increasing the spacing, considering that the established standard range has already demonstrated its capability to attain both normal mode and axial mode? If the spacing and radius are increased, it is expected that the size of the antenna will increase as well.

Response to R3C3: A major portion of the analysis executed in this paper is focused on identifying the sweet spot between the mechanical stability of the structure and its electromagnetic performance. To be able to achieve deployment of the QHA and conserve the same frequency of operation, the radius and spacing should change to maintain the same total length of the wire l , where the number of turns is $N=0.5$. So, when the radius of the wire increases, its spacing decreases and vice-versa.

As can be seen from the newly added Supplementary Figure 5, shown below, for normal and axial modes of a conventional QHA at the frequency of 1 GHz the total length of the wires differs. So, the same conventional antenna cannot operate in two different modes in the same frequency range.

Our proposed design was able to overcome this by preserving the same wire length for all the radii. This was achieved by modifying the spacing between the wires while taking into consideration the spacing-radius relation ensuring that the mechanical stability of the design is preserved as shown in Fig. 2(c) of the manuscript.

The figure below was added to the supplementary document as supplementary Figure 5.

Supplementary Figure 5: Total length of each wire of the conventional QHA at the frequency of 1 GHz for axial mode and normal mode in addition to our proposed structure

R3C4: Figure 2: In practical scenarios, how can the transition from State 1 to State 2 be implemented?

Response to R3C4: In practical scenarios, the transition from State 1 to State 2 and vice-versa is shown in Figure 7(f) in the revised manuscript (also shown below), where the fabricated antenna in its two states is presented. As can be seen, when a vertical downward force is applied to the tips of the QHA in stable state 1, the conductive strips (brown colored strips) will rotate clockwise around the joints, the non-conductive strips (white colored strips) will rotate counterclockwise around the joints, and the radius of the QHA antenna will increase while the height decreases. So, the antenna will transition to a stable state 2. On the other hand, as can be seen in the same figure, when a vertical upward force is applied to the tips of the QHA in stable state 2, the conductive strips will rotate counterclockwise around the joints, the non-conductive strips will rotate clockwise around the joints, and the radius of the antenna will decrease while the height increases. So, the antenna will return to stable state 1.

Figure 7 (f) in revised manuscript: State variation directions for the fabricated prototype

R3C5: Line 146 and Line 299 - Is there any empirical evidence or analysis available to support the claim that the proposed design exhibits low energy consumption?

Response to R3C5: Since the antenna is manually deployable and reconfigurable, it does not need additional electrical power for deployment including the use of actuators or external biasing to switch between different states. The only power needed is the power for the input of the transceiver, which is envisioned to be connected to the antenna as a back-end circuitry to ensure

communication. A potential transceiver is an off-the-shelf component from Analog Devices, discussed in (55) which requires 1.3 V and 2.9 mA for operation.

R3C6: Line 101 - 104 - Are there any figures available to support or substantiate those statements?

Response to R3C6: We updated **Supplementary Table 1** (also shown below) to support these statements. In fact, it was found that for a radius (R) less than 25 mm, the antenna structure is not well matched, and the maximum realized gain is negative. For $25 \text{ mm} < R < 35 \text{ mm}$, the antenna exhibits a close to omnidirectional pattern. A radius between $45 \text{ mm} < R < 55 \text{ mm}$ allows the antenna to exhibit a directional pattern with circularly polarized radiation behavior. For $R > 55 \text{ mm}$, the antenna design is no longer matched.

Supplementary Table 1: Antenna performance

Radius pec (mm)	Radius dielectric (mm)	S-parameter	Radiation pattern	AR	Frequency (GHz)	
60	60.6			CP	1.143	No matching
55	55.6			CP	1.115	Matching/ Directional radiation pattern
40	40.6			CP	1.094	Matching/ Transition state

35	35.6			NCP	1.087	Matching/ Transition state
30	30.6			NCP	1.08	Matching/ Omni- directional radiation pattern
25	25.6			NCP	1.073	Matching/ Omni- directional radiation pattern
20	20.6			NCP	1.0485	No matching/ Negative gain
10	10.6			NCP	0.9505	No matching/ Negative gain

We would like to thank Reviewer 3 again for taking the time to review the paper and provide feedback that helped us further improve the content of the paper and we hope that we have answered all the concerns that the reviewer presents.

REVIEWERS' COMMENTS

Reviewer #1 (Remarks to the Author):

Basically, I am not satisfied with the responses from the authors to my comments below.

'The proposed reconfigurable mechanism of the QHA seems interesting. However, the basic idea of this work was already presented in a major conference in this area [28].'

The authors have admitted that the basic concept, i.e. the principle of the work was already presented in their conference paper. The experimental verification and the conventional feeding network being implemented do not generate much novelty.

'For the application mentioned by the authors, a manually deployable QHA is not convenient. A few reconfigurable QHAs through electrical means were reported recently [a, b]. I would suggest the authors to add a mechanical actuation set-up to reconfigure the QHA through an electrical means and re-submit their work to an IEEE journal.

a. X. Yi, et al, 'Polarization and Pattern Reconfigurable Cuboid Quadrifilar Helical Antenna', IEEE TRANSACTIONS ON ANTENNAS AND PROPAGATION, VOL. 66, NO. 6, JUNE 2018.

b. W. Sun, et al, 'A Pattern Reconfigurable Circularly Polarized Quadrifilar Helix Antenna Through Phase Control', IEEE TRANSACTIONS ON ANTENNAS AND PROPAGATION, VOL. 70, NO. 9, SEPTEMBER 2022.'

The authors' suggestion on the additional electrical power consumption to operate the PIN diode or RF switches doesn't stand. The power for driving these electronic switches is in the order of a few - tens mW. For example, the PIN diode (Infineon BAR50-02L) in Ref [b] only needs about 50mW to operate. Also, the electronic switching time is very short, in the order of hundreds of nanosecond, and consumes very little energy. This energy burden is nominal in the application suggested by the authors, in which the transceiver needs a power supply in the order of tens - hundreds W. Of course, the electro-mechanical means proposed by the authors consume more energy and may add an additional burden to the power supply.

With the advancement of electronic technology, an electronic means for re-configuring an antenna is far superior than the mechanical or electro-mechanical ones. The proposed work may have some merits 30 or 40 years ago. It was the reason why I have suggested that the work is better to be evaluated in the IEEE antenna community.

Reviewer #2 (Remarks to the Author):

The revised version of the manuscript has highlighted clearly the contribution of this work to the field. I do not have any other comments.

Reviewer #3 (Remarks to the Author):

The authors have addressed the comments appropriately, but I have a concern about the following statement.

"To address this challenge, the reconfiguration mechanism in this work is made completely passive by relying on a manual reconfiguration process, thus removing the need for dc power and external

biasing"

The author intended to implement this antenna design in a disaster-prone area; does it make sense for them to risk their lives by reconfiguring the antenna manually? At least you have a backup power source to keep the device operational.

We sincerely thank the reviewers for their constructive feedback, which helped us improve the technical quality of the paper. We have carefully revised the paper to address the reviewers' comments. In this document, the comments are presented in blue followed by our point-by-point responses. All changes to the manuscript are colored in red in the revised manuscript document.

Reviewer 1 Comments:

R1C1: Basically, I am not satisfied with the responses from the authors to my comments below.

'The proposed reconfigurable mechanism of the QHA seems interesting. However, the basic idea of this work was already presented in a major conference in this area [28].'

The authors have admitted that the basic concept, i.e. the principle of the work was already presented in their conference paper. The experimental verification and the conventional feeding network being implemented do not generate much novelty.

Response to R1C1: We thank the reviewer for the comment. It is important to note that conference presentations serve as a platform to discuss novel ideas and concepts in a brief summarized manner to spark discussions, which is a clear, well-established norm in the scientific community. This practice does not infringe upon the comprehensive publication of a journal paper where intricate details of the design process, its novelty, contribution, and distinction are presented. In addition, in this manuscript, we have not only acknowledged this principle but also taken additional measures to underscore the significant contributions of our work beyond what is briefly discussed in the conference presentation. This was discussed in detail in our previous response and is summarized herein:

The novelty of the work resides in the merging of structural and electromagnetic perspectives from the initial design stage to the full implementation of the antenna system, its reconfigurable deployment mechanism, and its bi-stability along its fully dedicated feeding network all compose the novel aspect of this work. In addition, the comparison between any conventional QHA with our proposed topology and the design generalization with stability points prediction opens doors for future researchers to build upon this work to innovate further.

R1C2: 'For the application mentioned by the authors, a manually deployable QHA is not convenient. A few reconfigurable QHAs through electrical means were reported recently [a, b]. I would suggest the authors to add a mechanical actuation set-up to reconfigure the QHA through an electrical means and re-submit their work to an IEEE journal.
a. X. Yi, et al, 'Polarization and Pattern Reconfigurable Cuboid Quadrifilar Helical Antenna', IEEE TRANSACTIONS ON ANTENNAS AND PROPAGATION, VOL. 66, NO. 6, JUNE 2018.
b. W. Sun, et al, 'A Pattern Reconfigurable Circularly Polarized Quadrifilar Helix Antenna Through Phase Control', IEEE TRANSACTIONS ON ANTENNAS AND PROPAGATION, VOL. 70, NO. 9, SEPTEMBER 2022.'

The authors' suggestion on the additional electrical power consumption to operate the PIN diode or RF switches doesn't stand. The power for driving these electronic switches is in the order of a few - tens mW. For example, the PIN diode (Infineon BAR50-02L) in Ref [b] only needs about 50mW to operate. Also, the electronic switching time is very short, in the order of hundreds of nanosecond, and consumes very little energy. This energy burden is nominal in the application suggested by the authors, in which the transceiver needs a power supply in the order of tens -

hundreds W. Of course, the electro-mechanical means proposed by the authors consume more energy and may add an additional burden to the power supply.

With the advancement of electronic technology, an electronic means for re-configuring an antenna is far superior than the mechanical or electro-mechanical ones. The proposed work may have some merits 30 or 40 years ago. It was the reason why I have suggested that the work is better to be evaluated in the IEEE antenna community.

Response to R1C2: We agree with the reviewer that electrically reconfigurable antennas using switches constitute a powerful choice. However, this choice is not suitable for the application and the design perspective that we are tackling in this work. In fact, in this work, we rely on the bi-stability of the material composing the structure to ensure reconfiguration, a principle that guarantees passive reconfiguration mechanisms.

In addition, it is important to note that we are not proposing the use of any actuators in any way. We specifically indicated that the introduction of mechanical actuators falls beyond the scope of the work, which fully relies on the structure's bi-stability feature. We introduced the shape memory alloy actuator in our previous response to reply to the comment raised by the reviewer and to indicate that such an option is not valid in our scenario, even if it can automate the deployment process in its two states.

On the other hand, the deployment mechanism adopted for this structure eliminates the need to integrate PIN diodes or any other electronic switch into the feeding network or the antenna itself. Our proposed passive bistable deployment technique offers more robustness, and more immunity to environmental factors, and eliminates any nonlinearity effects that may result from the integrated switches. Furthermore, a bias-free system implicates less interference in the antenna's radiation characteristics, especially in a QHA topology which is an out-of-plane topology. Additionally, it is important to emphasize that realizing the unique reconfigurable functionality of our proposed QHA necessitates a mechanical modification of its structural topology, a feature that cannot be achieved using electronic switching reconfiguration.

We added a paragraph between Lines 339 and 347 to clarify this matter.

We would like to thank Reviewer 1 again for taking the time to review the paper and provide feedback that helped us further improve the content of the paper and we hope that we have answered all the concerns that the reviewer presents.

Reviewer 2 Comments:

The revised version of the manuscript has highlighted clearly the contribution of this work to the field. I do not have any other comments.

We would like to thank Reviewer 2 again for taking the time to review the paper and provide feedback that helped us further improve the content of the paper and we hope that we have answered all the concerns that the reviewer presents.

Reviewer 3 Comments:

R3C1: The authors have addressed the comments appropriately, but I have a concern about the following statement. "To address this challenge, the reconfiguration mechanism in this work is made completely passive by relying on a manual reconfiguration process, thus removing the need for dc power and external biasing"

The author intended to implement this antenna design in a disaster-prone area; does it make sense for them to risk their lives by reconfiguring the antenna manually? At least you have a backup power source to keep the device operational.

Response to R3C1:

We thank the reviewer for the detailed examination of the proposed operating scenarios. We agree that safety concerns can arise in certain disaster-struck areas that may affect the ability of the user to reconfigure the antenna. However, our overall intention is to deploy this antenna when needed for search and rescue in a disastrous situation. On the other hand, we note that in general low-infrastructure areas, bi-stability offers a low-power reconfiguration option for agile communications. If actuation is a must, then one interesting possibility is to have powered actuators as a backup, as noted by the reviewer. In this case, bi-stability is advantageous since the actuators need not be powered continuously and the shape will be maintained in the case of power loss to the actuators. **We add this consideration to our existing discussion about a powered vs. passive system in lines 326-328.**

We would like to thank Reviewer 3 again for taking the time to review the paper and provide feedback that helped us further improve the content of the paper and we hope that we have answered all the concerns that the reviewer presents.